# Are There General Features of How Immune Responses Are Regulated That Can Provide Clues to How Remitting/Relapsing Multiple Sclerosis May Be Treated?

**DOI:** 10.3390/ijms25052726

**Published:** 2024-02-27

**Authors:** Peter Alan Bretscher

**Affiliations:** Department of Biochemistry, Microbiology and Immunology, University of Saskatchewan, Heath Sciences Building, 105 Wiggins Road, Saskatoon, SK S5N 5E5, Canada; peter.bretscher@usask.ca

**Keywords:** antibody, B cells, multiple sclerosis, neuroimmunology, neurodegeneration, T cells, treatment

## Abstract

Most basic studies directed at how immune responses are regulated employ chemically “simple antigens”, usually purified proteins. The target antigens in many clinical situations, such as in autoimmunity, infectious diseases and cancer, are chemically “complex”, consisting of several distinct molecules, and they often are part of a replicating entity. We examine here the relationships between how immune responses to complex and simple antigens are regulated. This examination provides a context for considering how immune responses are regulated in those clinical situations involving complex antigens. I have proposed and discuss here a mechanism by which immune responses to the envisaged complex target antigen in remitting/relapsing multiple sclerosis go back and forth between inflammatory and non-inflammatory modes, potentially accounting for the course of this disease. This proposal makes predictions that can be tested by non-invasive means. It also leads to a suggestion for simple, non-invasive treatment.

## 1. Prologue

The immune system is centrally involved in many areas of medicine, primarily in autoimmunity, allergies, cancer, infectious diseases, and transplantation. There has been intense research directed in the last half century at delineating how immune responses are regulated. Such basic studies usually employ “simple antigens”, such as purified proteins, in order to achieve as elemental an experimental system as possible. The target antigen in many, if not most, clinical situations, in immune system-related areas of medicine, involve “complex antigens”, consisting of several chemically distinct molecules, and these are often part of a replicating entity. These observations naturally lead to the question of the pertinence of the basic studies on immune regulation to controlling immune responses involved in clinical situations. We discuss this very broad question here in a general fashion and in the context of remitting/relapsing multiple sclerosis (RRMS). We first consider the relationship between basic studies and immune responses to complex antigens. I envisage that this emphasis on basic studies, before I discuss the nature of immunity in MS, makes the exposition of my proposal more accessible. It allows the way my view of immune regulation differs from more conventional views to be apparent, and hopefully made plausible, before discussing its implications for MS. I argue that regulation of the class of immunity generated toward the complex antigens involved in MS is likely important in understanding the oscillating phases of this disease. I encapsulate my description in terms of Th1 and Th2 cells to simplify the exposition of my ideas. I later briefly address why I think this simplification can be accommodated with more general considerations. 

My hope in writing this article is that an individual, whose research is primarily on MS and who knows the currently dominant ideas as to how immune responses are regulated, but for whom these ideas are primarily accepted on trust, can gain from the story of how major ideas on immune regulation arose. I think we need such stories to transcend research silos.

## 2. Findings from Basic Studies on How Immune Responses Are Regulated

### 2.1. Variables of Immunization That Affect the Th1/Th2 of the Ensuing Response

One of the earliest studies examining how the dose of a simple antigen and time after antigen impact affect the nature of the ensuing immune response was undertaken by Salvin in the 1950s [1]. His findings are summarized in Figure 1. Low doses of antigen generate, in modern terminology, an exclusive Th1, delayed-type hypersensitivity (DTH) response. Moderate doses lead to a more rapid Th1 DTH response that declines as IgG antibody and Th2 cells are generated. The administration of an even higher dose of antigen results in even more rapid responses, and the Th1 DTH phase may be eclipsed. Salvin’s findings have been found to indicate how responses to diverse antigens are regulated, as I discuss later. 

### 2.2. Humoral Immune Deviation

Studies by a number of investigators in the 1960s showed that immunization, in a manner that results in the production of IgG antibody, renders the immunized animal refractory for the induction of a DTH response [2]. A DTH response is not generated upon immunization with an antigen challenge that generates such a response in naïve animals. It appears that the immune response is locked into an IgG humoral mode. This situation is referred to as a state of humoral immune deviation. Note that humoral immune deviation is likely pertinent to understanding why Salvin found that, upon immunization with a high dose of antigen, the expression of DTH declines as IgG antibody is produced; see Figure 1. 

### 2.3. Requirements to Establish and Maintain Self-Tolerance

Mitchison undertook an extensive investigation in the 1960s [3] that was well recognized forty to fifty years ago. I conclude from more recent conversations with my younger colleagues that most do not know of these classical studies, and from older colleagues that the significance of this investigation is not evident in terms of contemporary concepts. This investigation and related studies are consequentially largely forgotten. These studies are central to my thinking. I therefore think it helpful if I first trace the ideas inspiring, and summarize the observations that emanated from, these studies.

Burnet and Fenner had proposed in 1949 that tolerance to self-antigens was a consequence of their early presence in development, before birth [4]. Medawar and colleagues [5], as well as Hasek and his colleagues [6], showed that immunocompetent animals that had been exposed during development to a foreign antigen could no longer respond to this foreign antigen. These studies were correctly taken as evidence supporting Burnet and Fenner’s conjecture. It is technically difficult to administer an antigen to developing animals in most species. Immunologists therefore subsequently examined whether it was possible to generate unresponsiveness in neonates. This was possible in a number of cases. For example, Weigle undertook a series of such studies to which we shall return. It was natural in such circumstances to also pose the question of whether unresponsiveness could also be established in immunocompetent animals. This was particularly interesting once it was realized that tolerance is not uniquely established during development as originally envisaged by Burnet; it is an ongoing process throughout our life. This recognition came about in the following way. 

Most of the studies testing Burnet and Fenner’s conjecture were carried out with “foreign” stem cells that resulted in the adult animal being a chimera of self-cells and of cells derived from the foreign stem cells. Mitchison did a similar kind of experiment to Medawar’s but with non-replicating foreign cells [7]. He found that animals exposed during development were unable to respond to the target antigen when very young. However, they could respond at an older age unless the target antigen was periodically administered. This finding showed that tolerance was not uniquely and stably established in the developing animal. Maintaining tolerance was a continual process throughout life. Lederberg correctly recognized that this finding was consistent with the idea that lymphocytes are continually generated in adults [8]. Incidentally, this change in perspective also allowed scope for understanding how autoimmunity could arise in adults, as was known to often occur [9].

It is natural, if maintenance of tolerance is an ongoing process during adult life, to explore whether immunocompetent animals can also be made tolerant. In addition, self-antigens are usually present at a constant level, whereas the “concentration” of foreign antigens dramatically changes as the antigen impacts the immune system. I suspect but do not know that the virtual constancy in the level of self-antigens was a driving force in the design of Mitchison’s study of the mid-1960s. He gave immunocompetent mice a series of injections of the antigen over several weeks, with each mouse receiving the same dose each time but with mice belonging to different groups receiving different doses. The mice, after this regimen, were all given an antigen challenge that generated a robust IgG antibody response in aged-matched but naïve mice. He found that the IgG response to the challenge of the mice pre-exposed to the “priming” antigen fell into four groups, depending on the size of the priming dose, which varied over about a 10,000-fold range [3]. Mice injected only with saline exhibited a robust IgG antibody response. Those repeatedly pre-exposed to low, medium and high doses of antigen respectively generate a reduced, an enhanced and a reduced IgG antibody response to the challenge. Mitchison concluded that a virtually steady state of low and high levels of antigen reduced the IgG antibody response to the challenge by processes he respectively called low- and high-zone paralysis; a medium level of antigen primed the immune system for an antibody response. Mitchison interpreted the physiological significance of low- and high-zone paralysis as contributing to the mechanism of self-tolerance among immunocompetent lymphocytes [3]. I argue below for an alternative explanation.

### 2.4. Cell-Mediated Immune Deviation

Parish reported in the late 1960s a study of very similar design to Mitchison’s, employing again a simple antigen [10]. He used rats as his experimental animal. The major difference between the two studies was that Parish examined the state of DTH at the time of the antigen challenge. His observations confirmed and extended those of Mitchison. He found that the lower IgG antibody responses associated with “low-” and “high-zone paralysis” were associated with a state of DTH to the antigen, whereas there was no detectable or a lower level of DTH, respectively, in saline-injected rats or rats pre-exposed to medium doses of the antigen. Tolerance to self-antigens is a state of unresponsiveness for all classes of immunity. I suggest that Mitchison’s terms of “low-“ and “high-zone paralysis”, which reflect his idea that these processes reflect mechanisms of self-tolerance, is misleading. I refer to these states as low- and high-zone cell-mediated immune deviation, making clear their relationship with the processes controlling the class of immunity generated. Note the relationship between Salvin’s and Parish’s observations. Immunization once with a low dose of antigen to immunocompetent animals results in an exclusive DTH response, and immunization multiple times with a low dose of antigen results in a state of DTH and of cell-mediated immune deviation. 

### 2.5. Cellular Processes Involved in Immune Deviation

We showed in the mid- to late 1970s that mice immunized to produce IgG antibody harbor antigen-specific CD4 T cells that, when given to naïve mice, inhibit the generation of DTH in the recipient mice [11]. Conversely, we showed that mice in a state of cell-mediated immune deviation harbor antigen-specific CD8 T cells that inhibit, on transfer to naïve mice, the IgG antibody response of the recipient mice [11]. These in vivo studies preceded the discovery [12] by Mosmann and Coffman of Th1 and Th2 clones by several years. My theory of immune class regulation [13], which I briefly outline later, had predicted the general nature of our findings.

## 3. Coherence of Antibody Responses

When immunologists longitudinally examine the class/subclass of antibody present at a particular time after antigen impact, most of the antibody tends to belong to the same class/subclass. The class/subclass of antibody often evolves with time after antigen impact, but most of the antibody produced at one time is of a predominant class/subclass. For example, IgM antibody is produced before IgG antibody can be detected in primary responses and, most often, the production of IgM antibody decreases as substantial IgG antibody is produced [14]. This “coherence” of the antibody response is surprising in terms of some models of B-cell activation. The affinity for a simple antigen of antibody produced often varies over a 1000-fold range [3]. If the activation/differentiation fate of a responding B cell critically depends on the value of the affinity of its antibody receptors for antigens within this range, we would expect the differentiation fate of B cells, with heterogenous receptors, to be heterogeneous [14]. However, as outlined above, antibody responses appear to be coherently regulated. Such coherence, luckily for us (!), is readily explained by what we understand concerning the activation of B cells. Thus, all the diverse B cells specific for different epitopes of a simple antigen endocytose this same antigen and present the antigen’s diverse peptides. The differentiation fate of all these diverse B cells depends on the interleukin profile of the same population of activated Th cells specific for the nominal antigen. Thus, the coherence of the antibody response to a simple antigen is explained [14]. Consider now the antibody response to a complex antigen. An example of a “complex antigen”, much beloved by immunologists and employed in many classical studies, is sheep red blood cells (SRBCs) administered to mice [15]. It seems likely that there must be several chemically different components of this antigen that are foreign in mice and that these components are present in different amounts. The explanation for the coherence of the antibody response to simple antigens, outlined above, can also similarly account for the coherence of the antibody response to a non-replicating but complex antigens [14], such as SRBCs, as is known to occur [15]. 

## 4. Regulation of Immune Responses to Simple and Complex Antigens

It is known that infection with some fast-multiplying entities, such as some protozoa or bacteria, inevitably results in a rather rapid IgG antibody response. This finding can be accommodated with Salvin’s findings in that even infection with very low numbers rapidly results in what is effectively a high dose of antigen.

We have examined in mice the immune response to infection/injection with different numbers of relatively slowly multiplying entities, such as mycobacteria, the genus to which the pathogens causing leprosy and tuberculosis belong; *Leishmania major* protozoan parasites, responsible for the tropical disease of cutaneous leishmaniasis; and syngeneic, transplantable tumor cells [11]. In all cases, we found that infection with a sufficiently low number resulted in an exclusive Th1 response and with higher numbers in a response that evolved to have a substantial and usually a predominant Th2 component. Thus, all these complex antigens fit in with the generalizations that Salvin made upon immunizing once with a simple antigen; in particular, a sufficiently low dose of antigen or number of slowly growing entities generates an exclusive Th1 response. Moreover, we found that infection with a low number of these replicating entities, which resulted in a stable Th1 response, also resulted in a Th1 imprint; challenge with a higher number of the replicating entities, which in naive animals generates in time a response with a substantial Th2 component, resulted, in these pre-exposed animals, in a stable and large Th1 response [11]. Thus, all these complex antigens conform with Parish’s observation that sustained stimulation with low doses not only gives rise to Th1 responses but generates Th1 imprints.

## 5. Importance of Quantitative Considerations in Understanding the Regulation of Immune Class

Thus, diverse observations in diverse experimental systems support the importance of antigen dose in affecting the class of immunity generated. This support is even broader than just outlined. It is clearly of basic and practical importance to assess the breadth of the validity of such generalizations. Moreover, as these generalizations are of a quantitative nature, they bring quantitative considerations to the fore. There is another set of observations critical in further assessing the universality/non-universality of these generalizations. In addition, a further experimental generalization will be described that again illustrates the centrality of quantitative considerations.

We injected different strains of mice with different numbers of leishmania parasites and assessed the nature of the ensuing response. We could thereby define for each mouse strain a “transition number”, N_t_. Infection with a number of parasites below N_t_ results in a stable Th1 response, whereas infection with a number above N_t_ results in a response that in time develops a substantial Th2 component. Infection with a number considerably above N_t_ rapidly results in a predominant Th2 response. The value of N_t_ in different mouse strains varied over a 100,000-fold range [16]. This finding provides very strong evidence for the generality of the importance of low doses/numbers of slowly replicating entities in generating an exclusive Th1 response. It also indicates the importance of genetics in influencing the value of N_t_. 

As already documented, we have been able to generate exclusive Th1 responses by immunization with a low dose of antigen or infection with a low number of slowly replicating entities in diverse systems. Infection with very high doses or very high numbers, on the other hand, does not universally result in Th1 responses; in fact, it rarely leads to such a response. Parish’s high-zone cell-mediated immune deviation does not in practice hold for many antigens. However, the generality of the finding that stimulation with low levels of antigen generates exclusive Th1 responses means we have a general means for controlling whether an antigen generates a Th1 or Th2 response, unless the antigen is part of a rapidly replicating entity. This is important. Such an ability to control the Th1/Th2 phenotype of the response to most antigens is not evident in the context of the most popular frameworks that address how the Th1/Th2 phenotype of a response is determined, as I discuss below. 

Lastly, Pearson and Raffel proposed in the 1960s, based upon observation, that certain antigens were only able to generate (in modern terms) Th1 responses [17]. They identified such antigens as being minimally foreign, due either to their small size or being larger but only slight variations of a self-antigen. This is again a generalization where quantitative considerations are crucial. We tested a modified form of this proposal. We chose a low dose of a target antigen that generated an exclusive Th1 response. We examined what happened when we immunized with the same dose of the target antigen but conjugated to a foreign molecule. Immunization with the conjugate resulted in the increased generation of Th2 cells to the target antigen. Such modification of the response to the target antigen was much less evident in mice tolerant of the antigen conjugated to the target antigen [11]. These observations support Pearson’s and Raffel’s generalization. 

## 6. Models to Explain Immune Class Regulation

### 6.1. PAMP/DAMP-Centric Models

The most popular frameworks for what controls whether an antigen activates or inactivates its naive CD4 T cells is whether or not antigen impingement is associated with a pathogen-associated molecular pattern (PAMP) [18,19,20] or a danger-associated molecular pattern (DAMP) [21,22,23]. These models posit that a PAMP/DAMP-dependent signal is required to activate naïve CD4 T cells, and that an antigen can inactivate the CD4 T cells in the absence of such a signal. The grounds for these models have been well described by their proponents and are broadly known, so they will not be even outlined here. I have discussed extensively elsewhere why I find them implausible [9,11,14]. In addition, it is widely held that the nature of the PAMP/DAMP signal is of signature importance in determining the Th1/Th2 phenotype of the ensuing response [20,22,23]. This, too, I find implausible [11,14]. Although I have described the grounds for my scepticism elsewhere [11], I will briefly outline them here, in synoptic form, as these frameworks are so pertinent to the issues discussed.

We have reviewed above how important quantitative variables of immunization are in determining the Th1/Th2 phenotype of the ensuing response. The dependence on the dose of antigen, or number of slowly replicating entities, is true for foreign vertebrate antigens, such as SRBCs in mice, which are anticipated to be PAMP-free; for transplantable tumors; and for mycobacteria and protozoa, which express very different PAMPs. It seems a PAMP-independent mechanism is required to explain this dose dependence. Moreover, the Th1/Th2 phenotype of the response often evolves, after antigen impact, from an exclusive Th1 mode toward a Th2 mode. This is paradoxical to the PAMP/DAMP view, as the PAMP and DAMP signals do not change with time. Lastly, the alternative I favor, outlined in the next section, predicts that the partial depletion of CD4 T cells at the time of immunization will modulate a response with a substantial Th2 component toward a Th1 mode. This prediction has been tested by us and others in diverse systems, as reviewed in [11]. It is paradoxical to the PAMP/DAMP-centric view, as partial depletion of CD4 T cells is not anticipated to change the PAMP and DAMP signals. These considerations indicate why I think the PAMP/DAMP-centric views are incorrect and so an impediment to progress [11].

### 6.2. Threshold Hypothesis

This hypothesis was formulated in the early 1970s to describe the events that determined whether the activation of naïve CD4 T cells generates, in contemporary terms, Th1 or Th2 cells [13]. I then regarded, as one of its virtues, its ability to account for how all the known quantitative variables of immunization affect the Th1/Th2 phenotype of the ensuing response. I outline this feature below. I have recently described why I feel the hypothesis has become ever more plausible in view of the multiple tests of its unique predictions in diverse experimental systems [11]. I provide a synoptic account here to provide context. 

The two-signal model of lymphocyte activation was proposed as a minimal description of the activation and inactivation of mature lymphocytes that provides an explanation of peripheral tolerance, as outlined elsewhere [9]. This two-signal model provided the context for the formulation of the threshold hypothesis. The activation of CD4 T cells, according to the most recent formulation of the two-signal model, requires an antigen to facilitate the interaction of CD4 T cells mediated by B cells, as the antigen-resenting cells [11,13]. Many observations support this model [11]. The threshold hypothesis postulates that weak and robust CD4 T cell interactions give rise respectively to Th1 and Th2 cells. There are few CD4 T cells specific for minimally foreign antigens, and so, even in the presence of optimal doses of antigen to mediate CD4 T cell cooperation, only Th1 cells are generated. This hypothesis thus accounts for the Pearson and Raffel generalization described above. There are more CD4 T cells specific for more foreign antigens. In the presence of low amounts of antigen, the antigen-mediated CD4 T cell interactions will be weak and so result in the generation of Th1 cells. It is known that, after antigen impacts the immune system, helper T cells multiply, and so, as long as the level of antigen is sufficiently sustained, CD4 T cell cooperation will increase in intensity and the response will develop a Th2 component. Even greater amounts of antigen, more optimal for supporting CD4 T cell cooperation, will result in more rapid responses, and the Th1 phase may even be eclipsed. Thus, the proposed threshold mechanism accounts for the major generalizations of how quantitative variables of immunization affect the Th1/Th2 phenotype of the response [15]. The hypothesis predicts that partial depletion of CD4 T cells will modulate an immune response with a substantial Th2 component toward a Th1 mode. This “CD4 T-cell depletion” prediction has been tested and confirmed in diverse experimental systems [11] and, as discussed above, is difficult to square with the PAMP/DAMP-centric view [11].

### 6.3. Cytokine Milieu Hypothesis

The plausibility of a framework depends both upon what observations it can explain and whether there are other observations that appear incongruent. One cannot assess the plausibility of the threshold mechanism without addressing the multitude of observations showing the importance of cytokines in affecting the Th1/Th2 phenotype of the response and the further role of cytokines in whether Th cells belonging to other Th subsets are generated. I have recently addressed this issue [14], and so I just indicate here the view I favor.

I suggest there is more than one mechanism contributing to how the Th1/Th2 phenotype is determined. I propose the threshold mechanism is the primary mechanism. The envisaged role of cytokines is best introduced by a generalization about the activity of cytokines that Th cells produce. 

Most cytokines produced by Th cells that belong to one Th subset favor the further generation of Th cells belonging to this subset directly, or indirectly by inhibiting the generation of Th cells belonging to opposing subsets [14]. Examples would be the IL-4 made by Th2 cells that stimulates Th2 but not Th1 cells to multiply [24], and another the IFN-γ made by Th1 cells that inhibits the proliferation of Th2 but not of Th1 cells [25]. I argue that such properties of cytokines have interesting consequences; if a particular Th subset predominates in an antigen-specific population of Th cells, there is a tendency for this subset to become ever more dominant; in other words, Th cells of a particular Th subset tend to be self-promoting by virtue of the cytokines they produce [14]. 

Consider an immune response to a complex antigen. As its different components will in general exist in different amounts, they would, if uncoupled, generally induce immune responses of different Th1/Th2 phenotypes. The phenotype of the response to a prevalent component, p, would usually be different from that of a response to a scarce, other component, o. However, p and o will sometimes be either directly physically linked to one another or indirectly linked through another component of the complex antigen. Thus, an anti-o-specific B cell will not only present o-specific peptides but also other peptides generated by processing other components of the complex antigen to which o is linked. Thus, the strength of the cooperation determining the Th1/Th2 phenotype of an o-specific naïve CD4 T cell, according to the threshold mechanism, will depend not only on the number of other CD4 T cells specific for the nominal antigen o but also on the number of CD4 T cells specific for other components belonging to the complex antigen. In addition, the property of Th cells belonging to a particular subset to be self-promoting by virtue of the cytokines they produce contributes to making the Th response ever more coherent with time [14]. I provide two illustrative observations that support this cytokine implementation hypothesis. We showed that the IL-4 required to support the generation of Th2 cells is made by Th cells themselves [14]. Kelsoe showed that the coherence of the Th1/Th2 phenotype of immune responses increases as they evolve [26].

## 7. Circumstances Leading to Autoimmunity

Weigle’s studies of the 1960s [27,28], considered in a modern context, explain how autoimmunity can be induced by breaking peripheral tolerance [9]. In brief, the activation of naïve CD4 T cells requires CD4 T cell cooperation, whereas an antigen can inactivate single CD4 T cells. Naïve CD4 T cells specific for a peripheral self-antigen are inactivated, as generated one or a few at a time, by virtue of the continuous presence of the peripheral self-antigen. The impingement of a foreign antigen, F, that crossreacts with a peripheral self-antigen, pS, can induce the few CD4 T cells that are specific for both F and pS by virtue of there being more CD4 T cells specific for F than for pS. This mechanism explains how autoimmunity to peripheral self-antigens can be initiated, such as occurs upon infection by group A streptococci. An antigen of these bacteria crossreacts with an antigen of heart tissue, resulting in the activation of autoreactive CD4 T cells [29] and the production of autoantibodies [30]. This mechanism can also explain a most interesting feature of autoimmunity. The autoimmunity observed in the mouse model of autoimmune diabetes, and in the human disease itself, both display the phenomenon of “epitope spreading” [31]. This refers to the fact that the repertoire of the autoimmune CD4 T cells increases as the autoimmune response evolves; this population of Th cells responds to more and more different epitopes or peptide/MHC complexes with time. This phenomenon is naturally explained if the activation of CD4 T cells requires CD4 T-cell cooperation.

## 8. Basic Regulation of Immune Responses in the Context of Multiple Sclerosis

Firstly, it is not clear whether all destructive processes involved in different forms of MS or different stages of disease are mediated by the adaptive immune system [32]. However, it seems likely that the inflammatory response seen in the relapsing stages of RRMS is due to an adaptive immune response. Immune responses do not just disappear in the presence of antigen; it seems likely that remitting phases reflect a change in the class of immunity, as often observed during the course of immune responses. The remitting/relapsing phases of the disease are likely associated with inflammatory/less inflammatory modes.

There are three concepts I have tried to make plausible above that have been pivotal in my speculative thinking about multiple sclerosis.

Firstly, I have made a case for the general importance of antigen dose in affecting the class of immunity induced. This general dependency is not understandable within the PAMP/DAMP-centric view but is in terms of the threshold mechanism. I felt I had to make the case upfront for the importance of antigen dose, as it is central to the model I propose.

Secondly, the large majority of responses evolve in one direction. The most well recognized is the evolution of the response from a Th1 toward a Th2 mode, as outlined by Salvin [1]. Another is observed when individuals naturally grow out of allergies or during desensitization, when this can be successfully achieved as treatment [33,34]. These processes can be interpreted as a modulation of the immune response from a Th2 to “Th3” mode, associated in humans with the modulation of a mucosal response from the production of IgG_1_ and IgE antibodies toward the production of IgA and IgG_4_ antibodies [34]. I consider the modulation of immune responses against the “natural” trend of particular interest, as they may provide clues as to how the implied back-and-forth nature of the immune response occurs in RRMS.

I recognize two well-documented cases of immune responses being modulated in such a fashion. Visceral leishmaniasis is caused by a protozoan parasite. The pathogen is contained in individuals who generate a stable Th1 response, whereas patients have a Th2-like response at the time of diagnosis. Patients can be effectively treated by a short course of a few weeks of anti-parasitic drugs. This treatment results in a predominant Th1 response and so resistance to reinfection. Moreover, treatment is ineffective if the parasites are resistant to, i.e., not killed by, the drug. It seems most likely that this modulation of the immune response from a Th2-like toward a Th1 phenotype follows the killing of the parasites and so a lowering of the antigen load. This explanation is supported by the fact that treatment results in a change in the IgG isotypes among parasite-specific antibodies in a manner anticipated by such a modulation of the immune response [11]. Such a change in IgG isotypes, among the antigen-specific antibodies, is a highly practical way of longitudinally monitoring the Th1/Th2 phenotype of the response. We found it valuable in following the Th1/Th2 phenotype of immune responses in mice against tumors [11]. 

The second example of responses being modulated backward comes from a study of the responses of beekeepers to bee venom. The beekeepers exhibit a Th2 response, associated with predominant IgE production, upon receiving the first few stings of the season; the response rapidly switches to a T_reg-1_ mode, with the T_reg-1_ cells characteristically producing IL-10, and the predominant production of IgG_4_ antibody. This switch in the immune response results in a loss of IgE-mediated, immediate hypersensitivity to the bee venom [35]. It appears that, at the end of the season, with fewer or no stings, the beekeepers’ response switches to a Th2 memory mode, as evident by responses to the first few stings of the season. This again appears to be a case of backward modulation of the response following a lowering of the antigen load. Such modulation is again evident in the classes/subclasses of antibody produced [36]. 

Thirdly, recent and striking evidence shows that infection by Epstein–Barr virus (EBV) triggers the initiation of MS in some people [37]. Most individuals are infected by EBV, so, although such infection triggers MS in rare individuals, there are possibly other circumstantial variables, and certainly genetic variables, determining whether infection leads to MS. One genetic variable may be that in susceptible people, an EBV antigen crossreacts with a neural self-antigen, thus initiating an autoimmune response [38]. I think it plausible that such a trigger may initiate “epitope spreading” and in time Th responses to a collection of myelin antigens against which immune responses are known to be associated with MS [39]. I suggest these distinct chemical components collectively represent a complex antigen. 

## 9. A Model for RRMS and Implications for Treatment

A difference between immune responses to foreign and self-antigens is the effect of the response on the level of antigen that stimulates the response. In most cases, a response against a foreign antigen facilitates the removal of the antigen and, if this removal is sufficiently complete, the response decreases in size. In the case of responses to self-tissue, the autoimmune response may, in some cases, cause damage and the release of more antigen that stimulates the response. This idea is essential to the model I have proposed [39].

I think it helpful to first describe the model in its simplest form. Suppose the relapsing and remitting phases of RRMS are associated with predominant Th1 and predominant Th2 responses against the complex antigen. I also propose that the inflammatory Th1 response causes a greater release of components of the complex antigen than does the less inflammatory Th2 response. Consider the low level of antigen at the beginning of a relapsing phase, when a predominant Th1 response is favored. This response leads to an increased production of some of the antigenic components that are part of the complex antigen. The effective level of this antigen will therefore increase and may increase to a level where the response develops a substantial Th2 component with downregulation of the Th1 component. This modulation will in turn result in the release of lower levels of antigen and in time to substantially lower levels of antigen. The level may decrease to one where the Th2 component of the response is not sustained and the Th1 component is increased. This situation corresponds to the one first considered, and so the back-and-forth nature of the immune response and disease may be explained. 

This particular model critically depends upon the supposition that the generation of Th2 cells requires a higher level of antigen stimulation than does the generation of Th1 cells. A more general model can be envisaged in which the non-pathogenic and less inflammatory response is sustained by higher levels of antigen than the pathogenic and inflammatory mode associated with a relapse. There is not a consensus on what is the predominant type of Th cell involved in either relapsing or remitting phases, or even whether there is always a predominant Th subset [32]. I have therefore considered how the basic idea underlying the model in its most general form can be tested, as well as the therapeutic implications of the more general model. 

It is well established that the Th phenotype of an antigen-specific immune response affects the class/subclass of antibody produced against the antigen [12]. A prediction of the general model is that the class/subclass of antibody to the antigenic components of the complex antigen [35] will change with the phases of the disease. Testing this prediction does not involve any invasive procedures and so should be readily realizable.

A successful test of this prediction would, I suggest, provide sufficient grounds to examine whether the provision of the antigen just around or before a relapse would prevent a full relapse from occurring. It may also be that a longitudinal monitoring of the class/subclasss of antibody to myelin antigens may be useful in assessing when a relapse is imminent and so the envisaged treatment would be appropriate [39].

## 10. Conclusions

Our general understanding of how immune responses are regulated is consistent with a model of RRMS in which the autoimmune response can be in predominantly one of two modes. The inflammatory mode, associated with relapsing phases, results in the release and accumulation of considerable autoantigen, resulting in time in a transition to a less/non-inflammatory mode and remission. The non-inflammatory mode results in the release of less autoantigen and, in time, to a transition to the inflammatory mode and a relapsing phase. This model makes highly specific predictions that can be tested by non-invasive means. Should such tests be successful, it may be ethically justifiable to explore whether the administration of myelin antigens during a remission phase can sustain this phase and so avoid remission.

## Figures and Tables

**Figure 1 ijms-25-02726-f001:**
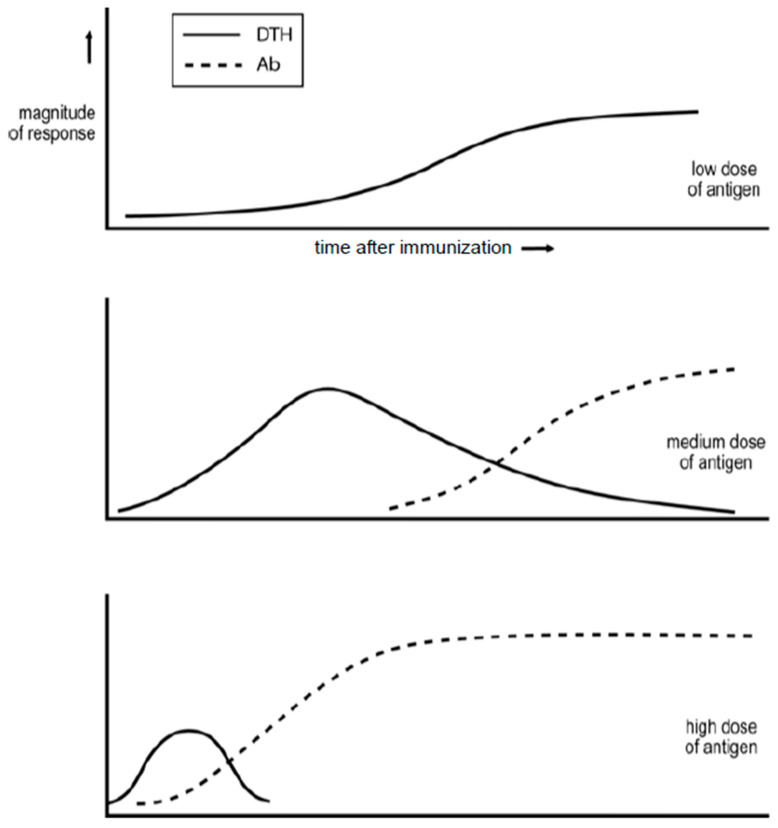
Synopsis of the observations of Salvin (3) on how the dose of a simple antigen and time after antigen impact affect the DTH/IgG antibody nature of the ensuing response.

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
