# Peer review of "Are There General Features of How Immune Responses Are Regulated That Can Provide Clues to How Remitting/Relapsing Multiple Sclerosis May Be Treated?"

_ijms, 2024, doi:10.3390/ijms25052726_

Round 1

Reviewer 1 Report

Comments and Suggestions for Authors

This a very nicely written narrative-type review. The author, a well-recognized scientist in the field of Immunology, drives us smoothly across the major hypotheses which have been put forward in the last 50 years or so in the context of immune tolerance. The author exposes his views on the concept of immune bias driven by the amount of antigens. Rather than the very nature of the targeted antigens or the patterns of associated DAMP/PAMP/cytokines, the antigen amount would be the main orchestrator of (auto-)immune responses. Relapsing multiple sclerosis is chosen as the disease which appears to fit best with this paradigm.

This is a very interesting and valuable work which may prove to be helpful in our understanding of MS pathophysiology. I see only one minor issue that the author may choose to address or not. While the quantitative parameter is nicely taken into account in this model, the “time” parameter appears to be neglected. Although the author quickly discusses the issue of repeated immune challenges, this is not integrated in this model. MS is a chronic disorder and it is highly likely that the impact of exerted by autoantigen amounts may change over time. Just because of the generation of memory lymphocytes.

In an ideal and complete model, since time goes hand in hand with space, the issue of the spatial compartmentalization of immune responses should be integrated too. But I guess AI would be needed to achieve such a goal.

Author Response

I thank the reviewer for his/her considered comments.

  The reviewer says:  "I see only one minor issue that the author may choose to address or not. While the quantitative parameter is nicely taken into account in this model, the “time” parameter appears to be neglected. Although the author quickly discusses the issue of repeated immune challenges, this is not integrated in this model. MS is a chronic disorder and it is highly likely that the impact of exerted by autoantigen amounts may change over time." I am a bit surprised by this comment. I try to make clear in the paper that the model proposed explains the different phases of the disease over time as due to the fact that the response in the inflammatory mode causes greater release of autoantigen than does the less/non-inflammatory mode. I realize the manuscript is long. I do hope a re-reading would allow the reviewer to agree with my response.

Reviewer 2 Report

Comments and Suggestions for Authors

The manuscript by Peter A Bretscher is not a classic review in its structure. Rather, this is a perspective or opinion article in which the author justifies his hypothesis about the mechanisms of the wave-like course of relapsing-remitting MS, based on the results of many years of his own and other studies. The hypothesis is very interesting, and the article should certainly be published.

The article is well written, however, in my opinion, it is overloaded with historical details (although they are interesting to read). Most of the article is devoted to the background of the issue (of course, this is important, but some historical details can be omitted) and a smaller part is devoted to the disclosure of the hypothesis regarding MS.

 In particular, other hypotheses related to the causes of the wave-like course of relapsed-remitted MS are practically not discussed. It would be good to devote one point to this.

 A few technical comments:

1) such a usual section as the Conclusion is not highlighted.

2) some final sections are missing, such as “Author contributions”, “Funding”, etc.

Author Response

I thank the reviewer for their careful comments.

    The reviewer refers to the fact that most of the manuscript deals with the history and justification of general concepts rather than observations more directly related to MS. This is correct. The reviewer felt this was not necessary. I have taken decades to get to the point where I lay out in a manuscript the justification of concepts that are central to the manuscript and inconsistent with the dominant frameworks of the day. I have found that without such justification, my proposals tend to be judged as implausible as being inconsistent with well accepted ideas. I also feel such general discussion of central issues is a valuable contribution. 

    I am not a specialist on MS. I have naturally read on the subject as I became interested in the subject and as the ideas outlined in the manuscript were developing. I have not come across clear models of what controls the relapsing/remitting transition. In view of the reviewer's suggestion that alternative views should be outlined and discussed, I searched for alternatives unsuccessfully. I would of course be willing to discuss them if provided with references where such alternative are proposed.  

        The technical points raised have been addressed.

Reviewer 3 Report

Comments and Suggestions for Authors

The manuscript presented here is well written and provides an interesting overview on MS.

I only have minor comments:

I would suggest developing more on recent findings. I understand that referring to the studies from the 60s and 70s is to point out where the current hypotheses come from, but nowadays, the field has seen tremendous developments, such as autoantibodies and the involvement of innate immune cells. I think you should expand on the innate immune system and how it communicates with adaptive immunity. I genuinely believe you should discuss this matter. Many articles show the intricated role of monocytes and microglia with T-B cells. In addition, depending on the Th response, microglia can react differently.

My last comment would be on the possible treatment. The title contains ‘That Can Provide Clues to How Remitting/Relapsing Multiple Sclerosis May Be Treated?’ but I don’t really see where you discuss how RRMS may be treated. It would be interesting to develop on that as well as the model which can be used. 

Thanks

Author Response

I thank the reviewer for their considered comments and openness to my describing so many older studies from the literature.

      The reviewer suggests the manuscript could be improved in two respects. The discussion should "expand on the innate immune system and how it communicates with adaptive immunity. I genuinely believe you should discuss this matter. Many articles show the intricate role of monocytes and microglia with T-B cells. In addition, depending on the Th response, microglia can react differently."  I feel I made a considerable effort to distil what I had to say to what was essential to the view developed. I tried to convey that I did not think the innate system does not affect the adaptive immune system, only that it does not do so in the particular ways proposed by Janeway and Matzinger. I have tried to limit my discussion to only what is essential. Unless really essential, descriptions of observations not pertinent to the issues at hand would make the already long article loose focus.

    The second point: "The title contains ‘That Can Provide Clues to How Remitting/Relapsing Multiple Sclerosis May Be Treated?’ but I don’t really see where you discuss how RRMS may be treated. It would be interesting to develop on that as well as the model which can be used." The final short  section explains the proposal and the rationale for treatment. I hope the reviewer would agree with men on a re-read. The idea is so simple after the framework is justified, that it only takes a line or two!   

Reviewer 4 Report

Comments and Suggestions for Authors

Are There General Features of How Immune Responses Are Regulated That Can Provide Clues to How Remitting/ Relapsing Multiple Sclerosis May Be Treated?

This review article examines the relationship between immune responses to complex and simple antigens. The author proposed and discussed a mechanism by which immune responses to the envisaged complex target antigen in remitting/relapsing multiple sclerosis go back and forth between inflammatory and non-inflammatory modes, potentially accounting for disease courses. This proposal makes predictions, which can be tested by non-invasive means, and provides suggestions for simple, non-invasive treatment. The review article is well-formulated and written like a storytelling. This review article helps readers develop basic concepts of immune responses. Following are the comments to further strengthen the current manuscript,

1.     Figure 1 warrants modification to make it of appropriate resolution. There is a background visible, and it does not look good.

2.     Consider providing the complete form of DTH (Delayed-type hypersensitivity) response on its first appearance.

Author Response

I thank the reviewer for their considered comments. I will try to improve Figure 1 and have spelled out that DTH refers to delayed-type hypersensitivity on its first appearance

Reviewer 5 Report

Comments and Suggestions for Authors

The paper presents an in-depth analysis of how immune responses are controlled, especially with respect to remitting/relapsing multiple sclerosis (RRMS). It covers key research on the regulation of immune responses, the effects of varying antigen amounts, differences in reactions to straightforward versus intricate antigens, and concepts like the Threshold Hypothesis and the Cytokine Milieu Hypothesis. This study proposes a framework for understanding the fluctuating behavior of RRMS, suggesting that variations in antigen quantities can sway Th1 and Th2 responses. Furthermore, it emphasizes the possibility of using antigen quantity adjustment as a therapeutic approach. Here are my comments:

-The Th1/Th2 paradigm has been foundational in immunology, but recent advances have expanded our understanding to include other T cell subsets (Th17, Treg) that play crucial roles in autoimmunity and MS. Any model that solely focuses on Th1/Th2 without considering these other subsets may not fully capture the disease's immunopathogenesis.

-Provided information in this review is not up-to-dated. Any model proposed for understanding RRMS must be critically assessed for its alignment with the latest clinical and experimental data. This includes understanding the role of various immune cells, cytokines, and how they interact within the CNS environment in the context of MS.

-It is written like a book chapter, not a research review. There should be more overviewed research studies, instead of reporting a lot of facts. 

Author Response

I thank the reviewer for their comments. I try to address their constructive comments below.

  1. The reviewer states: "The Th1/Th2 paradigm has been foundational in immunology, but recent advances have expanded our understanding to include other T cell subsets (Th17, Treg) that play crucial roles in autoimmunity and MS. Any model that solely focuses on Th1/Th2 without considering these other subsets may not fully capture the disease's immunopathogenesis." I agree with this statement. My response is that the role of dose of antigen is much ignored in the field. I document this for the Th1/Th2 paradigm, but also indicate its central role in responses involving other Th subsets (see discussion of responses to bee-stings.) My aim has not been to provide a comprehensive description of the immunopathogenesis of the disease in diverse patients, but to focus on what is essential to make a simple model plausible; this model is sufficiently substantial that it makes readily testable predictions that, if confirmed, may lead to tests of the effectiveness of a proposed therapy.  
  2. The reviewer suggests the information "in this review is not up-to-dated (sic). Any model proposed for understanding RRMS must be critically assessed for its alignment with the latest clinical and experimental data. This includes understanding the role of various immune cells, cytokines, and how they interact within the CNS environment in the context of MS." I actually disagree with this advice. I feel it is my duty to deal with any observations that are inconsistent with a model that is sufficiently substantial that it makes readily testable, non-invasive predictions, and, if these are confirmed, proposes a treatment. I think it essential to the value of this contribution that it is focused and does not attempt to be comprehensive in its description of diverse aspects of the field.
  3. The reviewer states: "It is written like a book chapter, not a research review. There should be more overviewed research studies, instead of reporting a lot of facts."  I hope the reviewer understands that I believe that some classical studies are crucially important but are ignored, as not readily understandable in terms of current paradigms. This perception has governed the style of this contribution. I am a bit surprised at the reviewer's comments. He/she summarizes the contents of the manuscript well. Given this, is not the critical to assess whether the model provides insight rather than the description of studies not directly related to the model?

Round 2

Reviewer 5 Report

Comments and Suggestions for Authors

My comments have not been addressed.